# Management of Oligoprogression in Patients with Metastatic NSCLC Harboring ALK Rearrangements

**DOI:** 10.3390/cancers14030718

**Published:** 2022-01-30

**Authors:** Chiara Pisano, Marco De Filippis, Francesca Jacobs, Silvia Novello, Maria Lucia Reale

**Affiliations:** Department of Oncology, S. Luigi Gonzaga Hospital, University of Turin, 10043 Orbassano, Italy; chiara.pisano@unito.it (C.P.); francesca.jacobs@edu.unito.it (F.J.); silvia.novello@unito.it (S.N.); realemarialucia@gmail.com (M.L.R.)

**Keywords:** oligoprogression, NSCLC, ALK rearrangement

## Abstract

**Simple Summary:**

The growing efficacy and availability of new targeted systemic therapies have markedly improved the prognosis of metastatic lung cancer patients harboring ALK rearrangements. The use of effective targeted therapies capable of maintaining a prolonged control of disease, for as long as possible, is paramount to ensure the best survival outcomes. In this regard, in cases of oligoprogression, “beyond progression” systemic treatment added to local ablative therapies is considered a feasible option in an attempt to improve the quality and quantity of patients’ lives, even if based on retrospective data. Certainly, treatment of ALK rearranged lung cancer patients with oligoprogressive disease must be individualized and based on multidisciplinary decisions. Above all, when further molecular targeted therapies are available, options must always be evaluated, especially in case of cerebral progression. In this review, we provide an updated and comprehensive overview of the main treatment strategies in cases of ALK rearranged oligoprogression.

**Abstract:**

Personalized treatment based on driver molecular alterations, such as ALK rearrangement, has revolutionized the therapeutic management of advanced oncogene-addicted NSCLC patients. Multiple effective ALK tyrosine kinase inhibitors (TKIs), with the amelioration of the activity at central nervous system level, are now available, leading to substantial prognosis improvement. The exposure to TKIs triggers resistance mechanisms and the sequential administration of other TKIs and chemotherapy is, for the most part, not targeted. In this context, extending the benefit deriving from precision medicine is paramount, above all, when disease progression occurs in a limited number of sites. Retrospective data indicate that, in oligoprogressive disease, targeted therapy beyond progression combined with definitive local treatment of the progressing site(s) is an effective alternative. In these cases, a multidisciplinary approach becomes essential for an integrated treatment strategy, depending on the site of disease progression, in order to improve not only survival, but also quality of life. In this review we provide an updated and comprehensive overview of the main treatment strategies in cases of ALK rearranged oligoprogression, including systemic treatment as well as local therapy, and report a real-world clinical story, with the final aim of identifying the most promising management for this subset of patients.

## 1. Introduction

Over the last two decades the application of personalized therapies based on molecular features has radically changed the diagnostic and therapeutic approach in advanced non-small cell lung cancer (NSCLC). Specific oncogenic drivers have been identified and several targeted drugs are now available, thus allowing for a significant improvement in terms of response rate, survival and quality of life in molecularly selected patients.

Epidermal growth factor receptor (EGFR) mutations, followed by anaplastic lymphoma kinase (ALK) rearrangements, are the pillars of precision medicine in thoracic oncology, being the first molecular targets identified with substantial clinical implications. 

ALK rearrangements are detected in 3–7% of advanced NSCLC and are typically associated with young age, non-smoking and adenocarcinoma histology [1]. ALK chromosomal rearrangements in NSCLC were first reported in 2007, and revolutionized the treatment of ALK+ patients. Several types of ALK translocation have been discovered, all resulting from the fusion between ALK gene portion encoding for the intracellular tyrosine kinase domain (exons 20–29) and different partners, of which echinoderm microtubule-associated protein like-4 (EML4) is involved in 90% of cases. This leads to the production of a chimeric protein with a constitutive tyrosine–kinase activity responsible for uncontrolled cellular proliferation [2]. Depending on the specific breaking point of the EML4 gene, more than 13 EML4-ALK fusion variants have been identified, with V1 (exons 1–13), V2 (exons 1–20) and V3 (exons 1–6) being the most common. A rapid bench to bedside approach from the detection of the ALK translocation in NSCLC has permitted the development of several effective targeted therapy options.

Under drug pressure, however, the emergence of molecular resistance finally occurs with a median time to disease progression from 10.9 to 34.8 months [3,4]. Sequential administration of ALK TKIs (from second to third generation) up to chemotherapy (platinum–pemetrexed) represents a common therapeutic approach. In this context, extending the benefit deriving from targeted treatment is paramount, above all when disease progression occurs in a limited number of sites, defining the so-called oligoprogressive state. Oligoprogression is, indeed, relatively common in oncogene-addicted NSCLC, involving about 30–50% of patients in the course of targeted treatment, a higher percentage if compared to patients treated with immunotherapy or chemotherapy [5], raising the crucial question regarding the best treatment approach: change of systemic treatment or treatment prosecution beyond progression combined with local ablative therapy.

In this review, we provide an updated and comprehensive overview of the main treatment strategies in cases of ALK rearranged NSCLC oligoprogression, including systemic treatment as well as local therapy. We report a real-world clinical story that supports the importance of multidisciplinary involvement and discussion in oncogene-addicted patients with an oligoprogressive disease. 

## 2. Oligoprogression in Oncogene-Addicted Disease: Focus on ALK-Rearranged NSCLC

Traditionally, metastatic disease has been regarded as incurable, regardless of number of metastases, with local therapy restricted to palliation of symptoms. However, this paradigm has recently been questioned. In 1995, Hellman and Weichselbaum first introduced the concept of oligometastatic disease (OMD), an intermediate state between localized and widespread tumor in which metastases are limited in number and location; compared with more advanced stages, OMD is more indolent and it is more amenable to ablative therapy for metastatic lesions [6]. Patients may present with de novo OMD (synchronous OMD) or with limited relapse after treatment with curative purpose (metachronous OMD). Ablative local treatment (e.g., radiotherapy, radiofrequency, crioablation and surgery) to all metastatic sites with radical intent has been shown to be potentially curative and associated with long-term survival benefit in several types of cancers, including NSCLC [7]. The precise definition of oligometastatic NSCLC is still a matter of debate in terms of number of lesions and sites involved. Studies conducted in oligometastatic NSCLC often used different cut-offs for number of metastases (generally not exceeding five), diverse eligibility criteria and various stadiation methods creating challenges in interpretation of clinical trial results [7]. To standardize inclusion criteria for future clinical trials, The European Organisation for Research and Treatment of Cancer (EORTC) Lung Cancer Group proposed a maximum of five metastases and three organs to be involved, without diffuse serosal metastases or bone marrow involvement, as long as all lesions are amenable to treatment with radical intent [7].

The concept of OMD led to the recognition of other disease states in which local ablative therapies (LATs) may have a potentially curative role. Oligopersistent or oligoresidual disease definition refers to patients who present with oligometastatic disease after an excellent response to systemic therapy [8]. Likewise, the definition of oligoprogressive disease was first introduced in 2012 to describe widespread systemic tumor at diagnosis, which shows, after a prolonged response or, at least, a stability of disease on systemic treatment, a limited progression in few metastatic sites and organs [9].

As mentioned above, oligoprogression is a common phenomenon in patients with oncogene-driven NSCLC treated with TKIs. Despite optimal initial response, progression eventually develops and, in roughly half of cases, relapse is limited to a few sites [9]. This pattern of progression is essentially explained by two mechanisms. (1) Pharmacokinetic failure: differently from next-generation TKIs, first-generation EGFR and ALK-TKIs have inadequate blood–brain barrier (BBB) penetration, hence isolated central nervous system (CNS) progression is easily detected [10,11]. (2) Intratumor and intertumor heterogeneity: resistant sites of disease may harbor a unique genomic profile, different from the primary tumor; drug pressure resulting from systemic targeted treatment prompts both the selection of intrinsically resistant cell clones and, after prolonged exposition, the development of acquired resistance mechanisms [12].

In contrast to widespread systemic progression, oligoprogression may be managed by not only changing systemic therapy, but also by maintaining the same systemic treatment beyond progression and adding metastasis-directed therapy to primary systemic treatment. Although most of the evidence derives from studies conducted in patients with EGFR-mutated NSCLC, the same approach can be extended to other molecular driver alterations, such as ALK, ROS1, BRAF and other less common genetic aberrations. Furthermore, this strategy can also be applied in non-mutated patients treated with immunotherapy alone or chemo–immunotherapy in order to exploit the abscopal effect and delay the transition to subsequent chemotherapy treatments known to be less effective in later lines. Given the prognostic and therapeutic implications, an accurate pre-treatment staging is required to define oligoprogressive disease and select those patients who will most likely benefit from the addition of aggressive local treatments. Compared with computed tomography (CT) alone, positron emission tomography/computed tomography (PET/CT) and brain magnetic resonance imaging (MRI) are recommended to achieve greater staging accuracy [13,14,15,16].

## 3. Local Ablative Therapies in Oligoprogressive ALK-Rearranged NSCLC

LATs to all progressing lesions in oligoprogressive, molecularly driven NSCLC is thought to potentially eradicate the TKI-resistant subclones, prevent secondary seeding and restore overall sensitivity of metastatic sites to the ongoing systemic treatment, until new resistant subclones emerge. Metastases-directed therapy approaches enable the extension in the duration of targeted therapy and delay the switch to the next systemic treatment as long as possible, thus prolonging the benefit of TKIs and improving survival. 

Stereotactic radiotherapy (SRT), an advanced radiotherapy technique with high local tumor control rates and low toxicity, is the most frequently adopted method in case of oligoprogressive, oncogene-driven NSCLC [17]. SRT includes stereotactic body radiation therapy (SBRT), which delivers high doses in divided portions, and stereotactic radiosurgery (SRS), which delivers a single ablative dose and is especially used for brain lesions. The addition of ALK-TKIs to radiotherapy can increase radiosensitivity of tumor cells strengthening the rational for combined strategies. Indeed, in preclinical studies, it has been shown that crizotinib elicits beneficial effects in combination with radiotherapy in ALK-positive NSCLC cell lines by reducing tumor proliferation, microvascular density and perfusion [18,19]. In another study, the antiproliferative effect of TAE684, a potent second generation ALK inhibitor that overcomes crizotinib resistance, was augmented by radiotherapy in EML4-ALK positive lung cancer cells [20]. On the other hand, another recent preclinical study reported that treatment with ALK-inhibitors combined with 10 Gy-irradiation led to similar effects to those of sole radiotherapy, with no clear evidence of sensitization to radiation by treating EML4-ALK mutated cells with ALK inhibitors [21]. 

Despite these preclinical heterogeneous data, clinical evidence, even if mostly retrospective, supports a metastases-directed therapy approach (Table 1). Several trials are ongoing in order to compare different treatment approaches for oligoprogressive disease (Table 2), and even if completing enrollment is not easy, it is now of paramount importance. Weickhardt and colleagues conducted a retrospective observational study to evaluate the benefits of LAT (radiotherapy or surgery) to oligoprogressive sites and the continuation of crizotinib or erlotinib in patients with metastatic ALK-rearranged (*n* = 15) or EGFR-mutant (*n* = 10) NSCLC, respectively. PFS1 was 9 months for patients treated with crizotinib and 13 months for patients treated with erlotinib. Patients with good ECOG PS (≤2) and non-leptomeningeal encephalic progression and/or ≤4 extra-CNS sites of progression were deemed suitable to receive LATs. A total of 25 out of 51 patients (49%) who progressed were judged suitable for local therapy (15 ALK+, 10 EGFR-MT), 24 patients received radiotherapy while one patient underwent surgery. Post-LAT, 19 out of 25 patients progressed again, with median PFS from the time of first progression (PFS2) at 6.2 months. In patients with isolated CNS progression, median PFS2 was at 7 months [9].

Two years later, the same group examined the safety and efficacy of LAT in ALK-rearranged NSCLC patients with extra-CNS (eCNS) disease oligoprogression during crizotinib therapy. Of the 38 patients analyzed, 33 experienced disease progression. Of these, 14 had eCNS progression suitable for radiotherapeutic LAT (<4 progressing lesions and ECOG performance status 0–1). Of note, one patient received adrenalectomy followed by an additional course of radiotherapy. Median PFS1 and median eCNS PFS1 in the 38 ALK+ patients were 9.1 months (1.1–41.1 months) and 17 months (1.1–34 months), respectively. Median duration of crizotinib exposure was 28 months in the LAT cohort versus 10.1 months for non-locally treated patients. Median PFS1 for patients receiving LAT was 14 months versus 7.2 months for those receiving only systemic therapy. For patients undergoing LAT, median eCNS PFS2 was 5.5 months (1–27 months) and local control rates at 6 and 12 months were 100% and 86%, respectively. The two-year OS in patients receiving more than 12 months of crizotinib was 72% versus 12% in those receiving less than one year of therapy (*p* < 0.0001). LAT was proven to be safe with an excellent toxicity profile; the only grade 2 side effect was fatigue, while no acute or delayed grade 3–5 side effects were observed [22].

Liu et al. reported the data derived from 38 ALK- or ROS1-positive patients treated with crizotinib “beyond progression” for a median period of 5.9 months. They found that median PFS2 was numerically longer among patients who received local therapy compared to those who did not receive it, even if the difference was not significant (9.9 versus 4.2 months, *p* = 0.094) [23].

In a recent multicenter retrospective study, Kroeze et al. assessed the efficacy of SRT concurrent to immunotherapy or targeted therapy in 108 NSCLC patients (60% on anti-EGFR/ALK TKIs). Fifty-six percent of patients were treated for oligoprogressive disease (5 or less metastatic lesions) with the brain being the most targeted site for SRT. Median PFS in oligoprogressive patients was 10.4 months. Second progression after LAT was once again oligoprogression in 59% of cases, regardless of the systemic treatment received (*p* = 0.765). After 1 year, 58% of patients were still receiving the same systemic treatment at the time of SRT. The median time to switch systemic treatment in oligoprogressive patients was 14 months (5.7–22.3 months). The following line was a new targeted therapy in 68% of cases [24].

In another recent multicenter retrospective study, Borghetti et al. evaluated the outcomes of 106 NSCLC patients with EGFR mutations and ALK rearrangements treated with radiotherapy concomitant to TKI. Fifty-two patients (42%) were defined as oligometastatic/oligoprogressive (up to four metastatic lesions). The most radio-treated sites were brain, bone and lung. Patterns analysis of disease recurrence after radiotherapy suggested that oligometastatic/oligoprogressive patients more frequently developed oligoprogression as subsequent tumor relapse. The 1- and 2-year OS rates in oligometastatic and oligoprogressive patients were 79% and 61.8%, respectively [25].

Ni et al. examined patterns of disease progression and the value of salvage radiotherapy in 93 ALK+ NSCLC patients treated with crizotinib. Progression rates at 12 and 24 months were 76.9% and 90.4%, respectively, with a median PFS1 of 11.5 months (95% CI; 9.8–13.2 months). Fifty percent of patients experienced disease progression at the original site, 23.1% developed metastases at sites not involved at the time of diagnosis and 26.9% of patients progressed in both sites. The brain was the most common site of disease progression, followed by lung and bone. Of the 52 patients with disease progression, 19 underwent salvage radiotherapy: 8 patients received whole brain radiotherapy (WBRT), 5 patients underwent brain SRS and 6 received extra-cranial SBRT. These patients achieved a higher PFS2 than those who did not receive salvage radiotherapy at crizotinib progression (10.0 vs. 6.0 months). In addition, patients who received any radiotherapy treatment during the course of disease had a longer OS than those treated exclusively with TKIs (*p* = 0.048) [26].

## 4. The Importance of Managing CNS Disease: Local Therapy and Systemic Treatment

ALK+ NSCLC is characterized by a marked neurotropism due to an intrinsic high affinity for nervous tissue of ALK rearranged cancer cells. Brain metastases are reported in approximately 26–40% of ALK rearranged patients at diagnosis, in 46–63% after a previous line of treatment with ALK-TKI and up to 75% after two or more ALK TKIs [28,29].

Isolated CNS progression with controlled extracranial disease is a common event on first-generation TKI treatment because of its reduced BBB penetration. Crizotinib activity on brain tissue is, indeed, modest with an ORR of 18% and a time to progression of 7 months [30]. During the first year of treatment with crizotinib, approximately one third of patients develop metastases to the brain, which represents the only site of progression for many patients. Consequently, the metastases-directed approach has been attempted several times in clinical practice in order to prolong TKI benefit. A small study retrospectively evaluated the continuation of crizotinib treatment after radiotherapy for seven patients with ALK-rearranged NSCLC and isolated CNS progression. Four patients received WBRT and three patients underwent SRS. PFS2 was 5.5 months with a range of 2.6 to 17.2 months [27]. 

In this context, SBRT and SRS have assumed a pivotal role in the multidisciplinary approach of oligoprogressive oncogene-driven NSCLC, with the aim of improving not only survival but also quality of life through a reduced impact on neurocognitive functions. Although SRS is routinely used for the treatment of a limited number of brain metastases, studies have shown that this technique can also be successfully used for the management of more than 10 metastases and limited disease burden, without increasing late neurocognitive toxicity [31]. Robin et al. evaluated 35 EGFR-mutated and ALK-rearranged NSCLC patients with multiple brain metastases (median number of lesions: 6; range: 4–26) who underwent SRS, even in multiple courses, comparing doses delivered to the hippocampus with SRS instead of WBRT or hippocampus-sparing WBRT. In patients treated with SRS on more than 10 brain metastases, the median dose delivered to the hippocampus and to the healthy brain was 1.2 Gy and 0.8 Gy, respectively, significantly lower than the doses delivered with a pancencephalic treatment, even if optimized to reduce the dose to the hippocampus itself. Median survival from diagnosis of encephalic metastases was 3.0 years (4.2 years for ALK-rearranged NSCLC and 2.4 years for EGRF-mutated NSCLC). Survival was comparable regardless of number of radiosurgery courses, number of brain metastases treated in total, or number of brain metastases treated in a single radiosurgery session. Mortality related to intracranial progression was 26% [32].

Before the advent of next-generation ALK TKIs, a metastases-directed approach was recommended after crizotinib failure in order to prolong TKI treatment period as long as possible. However, the role and the correct timing for LATs became less clear after the introduction of second and third generation ALK inhibitors, such as alectinib, ceritinib, brigatinib, ensartinib and lorlatinib, which have higher efficacy against common crizotinib resistance mutations and growing CNS activity. 

Analyzing patients with CNS progression during crizotinib treatment, Zhao et al. reported longer intracranial time to progression in patients who received sequential treatment with a second ALK TKI, including either ceritinib, alectinib, brigatinib or PLB1003, compared with those treated beyond progression after radiotherapy (*p* = 0.003) [33]. 

In the phase I ASCEND-1 trial, evaluating the safety and efficacy of ceritinib, 95 of the 255 patients enrolled had brain metastases at baseline. Of these, 19 patients were treatment naïve, 11 received ceritinib combined with radiotherapy and 8 received ceritinib alone. ORR (36.4% vs. 50%) and DCR (72.7% vs. 87.5%) showed no significant differences between the two treatment groups. The ASCEND-4 trial had similar results highlighting that radiotherapy did not add any additional benefit to TKI treatment (ORR 46.3% vs. 46.9%) [34,35]. Likewise, in a pooled analysis of two studies (NP28761 and NP28673) with ALK-positive NSCLC patients who had previously received crizotinib, alectinib combined with radiotherapy did not demonstrated a significant advantage compared to alectinib alone. CNS ORR was not statistically different between patients with prior radiotherapy (35.8%, *n* = 95) and patients without prior radiotherapy (58.5%, *n* = 41) [36]. 

Moreover, as previously discussed, Ni et al. reported that patients treated with LAT at crizotinib progression achieved a PFS2 of 10 months, comparable to or even lower than the PFS of patients progressing to crizotinib and treated with next-generation ALK inhibitors, such as ceritinib, alectinib, brigatinib and lorlatinib without LATs [26,37,38,39].

The “beyond progression” strategy and the use of LAT in cases of oligoprogressive disease also remain common with second generation ALK TKIs in a first-line setting, the current therapeutic standard. The landscape in which LAT is placed is characterized, however, by deeper intracranial efficacy and disease control because of the higher CNS penetration of newer generation ALK TKIs. In the global phase III ALEX trial, indeed, alectinib showed not only higher efficacy and better tolerability than crizotinib as first-line therapy, with a significant prolongation of median PFS (34.8 vs. 10.9 months; HR, 0.43; 95% CI, 0.32–0.58) but also an important CNS activity in terms of ORR (85.7% vs. 71.4%) and CNS progression incidence at 12 months (9.4% vs. 41.4%). Even if OS data are still immature, the 5-year OS rate is higher with alectinib (62.5% vs. 45.5%) [4,28,40].

Similarly, in the randomized phase III ALTA-1L trial, brigatinib demonstrated a longer median PFS (HR 0.48 [95% CI, 0.35 to 0.66]; log-rank *p* < 0.0001; median, 24.0 vs. 11.0 months) ORR (74% vs. 62%) and intracranial ORR (78% vs. 26%) compared to crizotinib, with superior tolerability and quality of life (QoL) [41]. More recently, in the eXalt3 trial, ensartinib demonstrated longer PFS (25.8 vs. 12.7 months; HR, 0.51 [95%CI, 0.35–0.72]; *p* < 0.0001), higher ORR (74% vs. 67%) and deeper intracranial RR (64% vs. 21%) compared with crizotinib [42].

Initially the third generation ALK TKI lorlatinib, characterized by high blood–brain barrier penetration [43] demonstrated important overall and CNS efficacy both in treatment-naïve patients and in patients progressing on crizotinib, second generation ALK TKIs, or both, with greater efficacy in patients harboring secondary ALK resistance mutations [27,44,45]. More recently in the CROWN study, comparing lorlatinib to crizotinib in patients with previously untreated advanced ALK-positive NSCLC, the 82% (95% CI, 57 to 96) of patients treated with lorlatinib and 23% of patients treated with crizotinib (95% CI, 5 to 54) had an intracranial response, respectively. At 12 months, the cumulative incidence of CNS progression as the first event was 3% with lorlatinib and 33% with crizotinib (HR, 0.06; 95% CI, 0.02 to 0.18). The percentage of patients who were alive without CNS progression at 12 months was 96% (95% CI, 91 to 98) with lorlatinib and 60% (95% CI, 49 to 69) with crizotinib (hazard ratio, 0.07; 95% CI, 0.03 to 0.17) [46]. 

Although radiotherapy is a cost-effective technique to achieve CNS disease control, delaying LATs is a valuable possibility with the advent of next-generation, brain-penetrating TKIs in order to ensure the best local control while preserving neurological function and quality of life. Aiming at optimizing decision-making in the management of CNS involvement, prognostic scales estimating survival for patients with brain metastases have been developed. The Graded Prognostic Assessment (GPA) score takes into account age, number of brain lesions, performance status and extent of extracranial disease to define prognosis, irrespective of tumor histology. This score has been updated, including the primary tumor as a prognostic variable, with the development of the Diagnosis-Specific GPA (DSGPA). The same group recently published the Lung-mol GPA, which considers lung cancer molecular profile for the presence of EGFR mutations and ALK rearrangements. Survival definition has been further improved, with a median survival of 47 months for the best prognostic group [47]. 

In general, because of the limited evidence available, in case of oligoprogressive disease, irrespective of sites of progression, inclusion in clinical trials is preferred.

## 5. Systemic Treatment Algorithm: Waiting for a Guide

As already discussed, considering the availability of further systemic treatment options is crucial to recognize the most effective and safe therapeutic strategy in oligoprogressive disease. However, as is well known, new generation ALK TKI selection and sequencing is still challenging because of the absence of head-to-head comparisons. In fact, second and third generation ALK TKIs, with an approximately 50% reduction in the risk of progression or death compared with crizotinib and with an impressive intracranial activity, overcame a first-line crizotinib approach (Table 3) [4,28,35,40,41,46,48].

In the absence of direct comparative trials, selection requires a comprehensive evaluation of drug systemic activity, CNS efficacy, drug interaction, safety and tolerability. Each ALK inhibitor has, indeed, its own unique side-effect profile with potential specific toxicities: diarrhea, nausea, and emesis with ceritinib; constipation and myalgias with alectinib; early-onset pneumonitis with brigatinib; rash, transaminase elevation, pruritus and nausea with ensartinib, altered lipid levels, cognitive and mood effects with lorlatinib [4,28,35,40,41,46,48].

All these factors are paramount to define the best systemic treatment strategy among all the available effective options.

Following “the-best-first” theory, lorlatinib could be considered as the preferred frontline TKI because of its higher activity in terms of PFS HR and CNS efficacy [5]. Preclinical and retrospective data demonstrated that sequential use of ALK TKIs could favor lorlatinib-resistant compound mutations, indicating that the most potent TKI used in first line treatment might better prevent acquired resistance mechanisms [49,50]. 

On the other hand, the sequence strategy, that keeps third generation TKI for subsequent lines, still remains a good option, considering the lorlatinib toxicity profile, the good activity and manageability of second generation ALK TKIs and the absence of targeted therapies available in cases of lorlatinib-resistance. In addition, frontline lorlatinib post-progression data are largely awaited since the acquired resistance mechanisms, paramount for defining the following treatment options, are not fully understood. In the attempt to define the most effective treatment algorithm, biomarkers predicting patients with potentially more aggressive disease or CNS-tropic ALK tumor biology would be helpful to better personalize a treatment strategy. For now, although ALK TKIs are approved regardless of any specific molecular alterations acquired at the time of disease progression, testing for resistance mutations is paramount to drive treatment choices because specific ALK resistance mechanisms can predict sensitivity to different ALK inhibitors. Intra-tyrosine acquired ALK mutations are the most common mechanism of resistance with second generation ALK TKIs, present in about 50% of cases. Among them, pG1202R is the most frequent, reported in more than 50% of patients, generally sensitive to third generation TKI [43]. Around 55% of patients who received lorlatinib developed two or more mutations, such as the most common L1196M/G1202R that confers high-level resistance to first, second and third generation ALK TKIs, supporting the concept of progressive stepwise genetic complexity due to the selective pressure of sequential TKIs [50]. In the other cases, the development of acquired resistance is caused by off-target mechanisms, such as bypass signaling, histological and/or phenotypical transformations [43,51]. A phase II prospective MASTER protocol (NCT03737994) is currently evaluating a genomic-driven therapeutic sequence in ALK-rearranged metastatic NSCLC who progressed on prior second generation ALK TKIs, based on tumor genotype at the time of disease progression [52]. Indeed, re-biopsy and molecular profiling could be particularly helpful to define a personalized treatment algorithm allowing for the detection of resistance mechanisms and of sensitive alterations as well as allowing for the identification of genomic-driven therapeutic sequences. This information could be of crucial importance, above all when the sites of progression are limited and the need to change systemic treatment or to associate a local therapy is largely debatable [51].

When ALK TKI options are exhausted (e.g., in lorlatinib-refractory patients), first and second generation ALK TKI are not effective, apart from rare and exceptional cases. New targeted therapies are currently not available. Some promising molecules have been studied but results are still immature [53]. 

Platinum-based chemotherapy, mainly with pemetrexed, is still a valuable option of treatment [54]. The combination of carboplatin–paclitaxel–bevacizumab and atezolizumab represents, for now, the only treatment regimen with an immune checkpoint inhibitor demonstrating a significant efficacy in oncogene-addicted patients with objective responses >50% [55].

## 6. Clinical Case 

### 6.1. Patient Information and Presentation

Patient XX, a 56-years-old female (Figure 1). 

Past medical history: systemic arterial hypertension, no first-degree history of cancer.

Smoking history: former light smoker.

Presentation: patient presented to the oncology department in May 2011 after occasional finding of right pleural effusion during pre-surgical tests for thyroid nodes, general good clinical conditions (ECOG PS 0). 

### 6.2. Metastatic NSCLC Diagnosis

Contrast CT scan and 18FDG-PET/CT confirmed right pleural effusion and documented the presence of middle lobe mass, multiple right lower lobe nodes, pleural and mediastinal lymph nodes involvement. Histological examination of pleural biopsy: NSCLC, adenocarcinoma histotype. Biomarker tests: EGFR wild-type; EML4-ALK rearrangement.

### 6.3. Patient Journey

Because of the absence of approved targeted treatment at that time, from October 2011 to February 2012, patient XX was treated with first-line cisplatin 75 mg/m^2^ + pemetrexed 500 mg/m^2^ (6 cycles). Best response (BR): stable disease. In July 2013 thoracic and abdominal disease progression occurred (increase of lung parenchymal lesions, carcinomatous lymphangitis, extension in pleural involvement, axillary and abdominal lymphadenopathies). Patient XX was started on ceritinib, taking part in a clinical trial. BR: partial response of disease. In May 2015 the CT scan recorded a new single encephalic lesion at right temporal lobe. Multidisciplinary evaluation: in consideration of the oligoprogressive pattern, the patient underwent exeresis of the right temporal lesion while maintaining ceritinib treatment. Histologic report of brain tissue: lung adenocarcinoma. Biomarker tests: EML4-ALK rearrangement, no evidence of ALK resistant mutations. Treatment with ceritinib, initially well tolerated, was definitively discontinued in April 2016, after progressive dosage reduction, for refractory G3 transaminases elevation. The patient continued with close clinical and radiological monitoring. In July 2016, because of a widespread encephalic progression (new left frontal lobe and left parietal lobe lesions, dimensional increase of previously treated right temporal, left parietal and right parietal lobe metastases) WBRT was performed (30 Gy/10 fr), after multidisciplinary evaluation, with a good radiological and clinical response to treatment. A few months later (October 2016), because of abdominal lymph nodal progression of disease, a new systemic treatment with brigatinib 180 mg/day in the Early Access Program (EAP) was prescribed. BR: partial response (regression of almost all encephalic lesions and partial regression of abdominal lymph nodes). In June 2018, after almost two years from the start of treatment, because of the appearance of a new left frontal–parietal metastasis and the increase of the right temporal one, SRS was performed (18 Gy each). Disease remained stable until October 2020 when a new disseminated encephalic progression occurred. The patient was still in good clinical condition, ECOG PS 1, with only mild cognitive impairment after WBRT. Fourth-line treatment with lorlatinib 100 mg/day in EAP was started.

Patient XX is alive and lorlatinib treatment is ongoing (last follow up in November 2021). This 10-year-long story is an illustrative example of how proper management of oligoprogression can prolong TKIs benefit in oncogene-addicted NSCLC, thus significantly improving survival and quality of life.

## 7. Perspectives and Conclusions

Altogether, these data confirm the success of treatment development for ALK-rearranged NSCLC patients and the fast translation of these progresses into remarkable clinical outcome improvements. At the same time, they raise the questions about how to maximize the benefit potentially deriving from the expanding plethora of effective options (systemic and local ones). Retrospective results certainly justify the attempt of prolonging the benefit of targeted therapies by associating local treatment in cases of oligoprogression, above all when no more targeted therapies are available. The volume and location of the progressive site, the toxicities related to the different approaches (systemic therapies or ablative ones), patient’s general conditions, age, comorbidities, and, finally, multidisciplinary discussion, are undoubtedly paramount in the decision, particularly in a setting in which prospective data are lacking [56]. The results of ongoing clinical trials are eagerly awaited (Table 2). Indeed, these results, together with the identification of specific biomarkers predicting benefit from local therapy or widespread progression, are potentially the key to recognize the best treatment approach for every oligoprogressive ALK rearranged lung cancer patient.

## Figures and Tables

**Figure 1 cancers-14-00718-f001:**
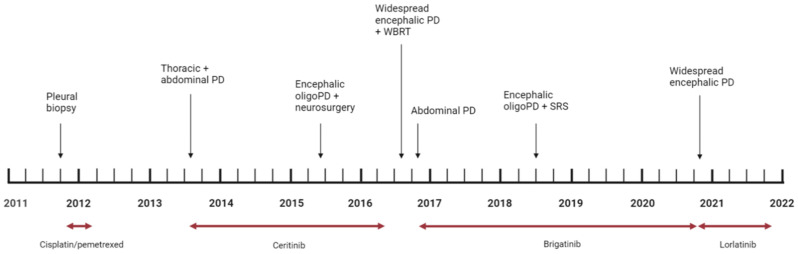
Patient XX oncological history. PD: progressive disease; oligoPD: oligoprogression; WBRT: whole brain radiotherapy; SRS: stereotactic radiosurgery.

**Table 1 cancers-14-00718-t001:** Selected trials of oligoprogressive NSCLC patients treated with local ablative therapies combined with TKI.

Author (Year)	Type of Study	N. of Patients	Molecular Status	LAT	Sites	mPFS (Months)	mOS (Months)
Weickhardt et al. (2012) [9]	Retrospective	25	EGFR+, ALK+ (54%)	SBRT, SRS, WBRT, XRT, surgery	CNS + eCNS	6.2 *	-
Gan et al. (2014) [22]	Retrospective	14	ALK+ (100%)	SBRT, HRT, surgery	eCNS	5.5 *	39
Liu et al. (2018) [23]	Retrospective	38	ALK+ (86.8%), ROS1+	SBRT, WBRT	CNS + eCNS	9.9 *	-
Kroeze et al. (2021) [24]	Retrospective	108	EGFR+, ALK+ (15%), ROS1+, WT	SRS, SBRT	CNS + eCNS	10.4	-
Borghetti P et al. (2019) [25]	Retrospective	106	EGFR+, ALK+ (19%)	SRT, HRT	CNS + eCNS	-	23
Ni et al. (2019) [26]	Retrospective	19	ALK+ (100%)	SBRT, SRS, WBRT	CNS + eCNS	10 *	-
Takeda et al. (2013) [27]	Retrospective	7	ALK+ (100%)	WBRT, SRS	CNS	5.5 *	-

LAT: local ablative therapy; mPFS: median progression-free survival; mOS: median overall survival; TKI: tyrosine kinase inhibitor; HRT: hypofractionated radiotherapy; SBRT: stereotactic body radiation therapy; SRS: stereotactic radiosurgery; XRT: standard radiation therapy; SRT: stereotactic radiotherapy; WT: wild-type; CNS: central nervous system; eCNS: extra-CNS. * Calculated from progression on TKI. The Kroesze et al. study included both oligoprogressive (≤5 metastatic sites) and polyprogressive (>5 metastatic sites) patients. In oligoprogressive patients (56%) mPFS was 10.4 months. The Ni et al. study included both oligoprogressive/oligometastatic (≤4 metastatic sites) and polyprogressive (>5 metastatic sites) patients. Fifty-two patients were defined as oligometastatic/oligoprogressive. SRT with an ablative intent was administered in 49 patients.

**Table 2 cancers-14-00718-t002:** Main ongoing clinical trials in oligoprogressive NSCLC.

Trial	Active Comparator Arm	Experimental Arm	Tumor Type	Molecular Selection	Definition of Oligroprogression	Primary Endpoint	Phase	Status
NCT02756793 (STOP trial)	SoC (Continue with current systemic agent(s)/Observation/Switch to next-line treatment	SABR + continuation of current systemic agents	NSCLC	-	Maximum of 3 progressing metastases in any single organ system and the total number of metastases must be 5 or less	PFS	-	Active, not recruiting
NCT04519983	-	Upfront TKI + Salvage SRT	NSCLC	EGFR	Intracranial oligo-progression	iORR	2	Not yet recruiting
NCT04405401 (SUPPRESS-NSCLC)	SoC (Switch to subsequent systemic therapy line/BSC/continue current systemic line)	SABR to oligoprogressive lesions + continue current systemic therapy	NSCLC	-	1–5 extracranial lesions	PFS, OS	2	Recruiting
NCT03256981 (HALT)	Continued TKI therapy alone	SBRT and continued TKI therapy	NSCLC	actionable mutation treated with TKI	≤3 extracranial sites of progressive disease	PFS	2/3	Recruiting
NCT03808662	SoC	SBRT	NSCLC, breast cancer	Non oncogene-addicted cohort + oncogene-addicted cohort	1–5 oligo-progressive lesions	PFS	2	Recruiting

SoC: standard of care, NSCLC: non-small cell lung cancer, PFS: progression-free survival, iORR: intracranial objective response rate; BSC: best supportive care; OS: overall survival.

**Table 3 cancers-14-00718-t003:** Comparison of first-line phase III trials in advanced ALK-rearranged NSCLC patients.

	PROFILE 1014 [3]		ALEX [11]		ALTA-1L [12]		eXalt3 [13]		ASCEND-4 [15]		CROWN [22]	
	Crizotinib	Chemotherapy	Alectinib	Crizotinib	Brigatinib	Crizotinib	Ensartinib	Crizotinib	Ceritinib	Chemotherapy	Lorlatinib	Crizotinib
ORR	74%	45%	83%	76%	74%	62%	74%	67%	72.5%	26.7%	76%	58%
mPFS (months)	10.9	7	34.8	10.9	29.4	9.2	25.8	12.7	16·6	8.1	NR	9.3
PFS HR	0.45; 95% CI, 0.35–0.60		0.43; 95% CI 0.32–0.48		0.43; 95% CI, 0.31–0.61		0.51; 95% CI, 0.35–0.72		0·55; 95% CI 0·42–0·73		0.28; 95% CI, 0.19–0.41	
Intracranial response rate	-	-	81%	50%	78%	26%	63.6%	21.1%	72·7%	27.3%	82%	23%
Frequency of dose reduction	-	-	20%	20%	38%	25%	24%	20%	80%	45%	21%	15%
Frequency of discontinuation	12%	14%	15%	15%	13%	9%	9%	7%	5%	11%	7%	9%
Key adverse events	dision disorders, diarrhea, nausea, and edema	nausea, fatigue, vomiting, and decreased appetite	myalgia, increased blood bilirubin, increased ALT	nausea, diarrhea, and vomiting	ILD/pneumonitis, nausea, CPK increase, ALT increase, lipase increase	nausea, diarrhea, edema	Rash, pruritus, nausea, pyrexia, ALT and AST increase	liver toxic effects, nausea, edema, and constipation	diarrhea, nausea, vomiting, AST and ALT increase	nausea, vomiting and anemia	Edema, Hypercholesterolemia, hypertriglyceridemia, increased weight, neuropathy, cognitive effects, speech effects	Diarrhea, vision disorder, vomiting, increased alanine aminotransferase level, fatigue, constipation, increased aspartate aminotransferase level, decreased appetite, dysgeusia, and bradycardia

ORR, objective response rate; PFS, progression-free survival; NR, not reached; CNS, central nervous system; HR, hazard ratio; CI, confidence interval.

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
