# Peer review of "Management of Oligoprogression in Patients with Metastatic NSCLC Harboring ALK Rearrangements"

_cancers, 2022, doi:10.3390/cancers14030718_

Round 1

Reviewer 1 Report

Chiara Pisano et al. review in their article the treatment options and perspectives of patients with oligoprogressive metastatic non-small-cell lung carcinoma (NSCLC). 

The treatment of NSCLC patients is mostly carried by first, second and third generation tyrosine kinase inhibitor and local ablative therapy (mostly radiotherapy but in selected cases operational extraction as well) of selected metastases. Even though NSCLC is an aggressive disease with a very poor prognosis when only first generation tyrosine kinase inhibitors were available, today's treatment can prolong the patient's life for many years with a very good life quality. Local ablative therapy is overall less effective than medical treatment with third generation tyrosine kinase inhibitors. It is still indicated when the tumor becomes drug resistance, probably by clonal selection. 

The article gives a good overview of the current treatment options for NSCLC patients, citing a lot of studies involving different drugs. All studies are retrospective and lack a control group which is understandable considering the aggressiveness of the disease. The tables give a good overview of the results of the different treatments and the type of side effects. 

I think the article can be published as it is. 

Author Response

We thank the Reviewer for the comment.

We hope that this review can be useful to the reader interested in an overview of treatment strategies in oligoprogressive ALK- rearranged NSCLC. We agree that, despite the improvements in this setting, available studies are, for now, retrospective and heterogeneous. Prospective data are eagerly awaited in order to define the best treatment algorithm. Thank you again for your revision.

Reviewer 2 Report

In order to benefit deriving from ALK-TKI, local treatment for a limited number of sites may be useful. Authors provided an updated and comprehensive overview of the main treatment strategies in case of ALK rearranged oligoprogression, including systemic treatment as well as local therapy, and report real-world clinical stories, with the final aim of identifying the most promising management

This review seems to be beneficial for pulmonary oncologists. I have a few comments.

Authors well reviewed about oligoprogression in patients with metastatic NSCLC harboring ALK rearrangements and showed some data concerning EGFR mutants together. Are there differences of treatment strategy for oligoprogression among patients with the different oncogene drivers (EGFR, ALK, or ROS1, etc)? Moreover, is treatment strategy for oligoprogression while using molecular targeting agents different from chemotherapeutic agents and immune checkpoint inhibitors? Please discuss above.

line 399 & 439: What does LMT mean?

Author Response

We thank the Reviewer for the comment and for the interesting points of reflection. The strategy of continuing the same anticancer agents beyond progressive disease after local ablative therapy is common in NSCLC management. In patients harbouring driver mutations this approach, even if supported by retrospective data, are recommended in different guidelines, regardless of the driver alteration. As expected, data are larger in EGFR + population. The prognosis in oligoprogression during the administration of immunotherapy is promising and biological intriguing. Concerning cytotoxic chemotherapy, the attitude seems to be less univocal (data are lacking also in this subgroup). However LAT and subsequent continuation of the current chemotherapeutic drugs beyond PD is a common habit in presence of substantial clinical benefit above all when alternative options are limited.  We discussed these aspects in the text.

line 399 & 439: What does LMT mean?

LMT was patient’s name, we replaced it with “patient XX” to avoid misunderstandings.

We thank the Reviewer for the precious suggestions.

Reviewer 3 Report

This was a potentially  good review on management of lung cancer patients with oligoprogression after initial good response or stable disease on initial treatment.

Looking at the title, it gave the impression that the content would be focused on ALK rearrangement. However, a lot of details on other driver mutations like EGFR were discussed and included.

The authors need to decide whether to focus on ALK to match with the title or the title of the article should be refined in include other driver mutations not limited to ALK.

It's also notable that within the discussion, the authors have not separate the issue of oligo-progression in brain / CNS (esp common for ALK tumors) vs oligo-progression at other sites, that would affect management and hence prognosis.

Major concerns:

  1. Page 2, line 45, EGFR but not ALK was the first molecular target....
  2. Page 2, line 55 - 57. Please elaborate on how experimental approach has led to development of several targeted therapeutic options.
  3. Page 2, line 59 - 61. Please explain how sequential administration of ALK-TKI up to chemotherapy represents a common approach.
  4. Page 2, line 71, the two real-world stories 'support' (better than to say 'prove' the importance of ......
  5. Page 3, line 140, should be high doses in 'divided portions'
  6. Page 3, line 143, suggest to replace 'compound' with 'combined' strategies.
  7. Page 6, line 265, please specify which is (are) the 'innovative' ALK inhibitors?
  8. Page 6, line 275, please specify ASCEND-4 trial gave 'similar results' in what sense?
  9. Page 6, line 286, please specify which 2nd generation ALK-TKI is being referred to here?
  10. Page 7, line 314 - 317, cannot understand the phrase 'These data of important central efficacy highlight that, .........
  11. Page 7, line 337, please elaborate on what was meant by putting crizotinib on the corner?
  12. Table 2, please define what is frontline?
  13. Page 9, line 381, please elaborate on how re-biopsy and molecular profiling will guide subsequent treatment, in case of ALK tumor, or tumors with other targets?
  14. Page 9, line 384 - 386, sentence meaning unclear, please elaborate further or re-write this part.

Minor concerns:

  1. Page 3, line 115, Different from .......
  2. Page 5, line 189, LAT 'was proven' to be safe......
  3. Page 6, line 269, please unify style of reporting p values.
  4. Page 9, line 345, each TKI has, indeed, 'its own unique' side effect profile.....
  5. Figure 1, 'addominal' should be 'abdominal'
  6. Line 413, use SI unit and standard abbreviations for 'mq' and same for subsequent appearance of 'mq'
  7. Line 430, /die?

Author Response

We thank the Reviewer for the comments. The text is mainly focused on ALK-rearranged NSCLC management, however because available data about  oligoprogessive setting are limited and the major studies in this setting included both EGFR and ALK+ subgroups, EGFR population has also been mentioned. We agree with the importance of managing CNS disease, we specifically dedicated a paragraph.We implemented/modified our manuscript following the suggestions in order to improve the clarity of our paper.

Major concerns:

  1. Page 2, line 45, EGFR but not ALK was the first molecular target....

Thank you for the comment, we clarified the concept.

  1. Page 2, line 55 - 57. Please elaborate on how experimental approach has led to development of several targeted therapeutic options.

Thank you for the comment, we implemented the concept.

  1. Page 2, line 59 - 61. Please explain how sequential administration of ALK-TKI up to chemotherapy represents a common approach.

Thank you for the comment, we clarified the concept where indicated and improved the paragraph specifically dedicated (Systemic treatment algorithm: waiting for a guide)

  1. Page 2, line 71, the two real-world stories 'support' (better than to say 'prove' the importance of ......

Thank you, modified.

  1. Page 3, line 140, should be high doses in 'divided portions'

Thank you, modified.

  1. Page 3, line 143, suggest to replace 'compound' with 'combined' strategies.

Thank you, modified.

  1. Page 6, line 265, please specify which is (are) the 'innovative' ALK inhibitors?

Thank you, specified.

  1. Page 6, line 275, please specify ASCEND-4 trial gave 'similar results' in what sense?

Thank you, specified.

  1. Page 6, line 286, please specify which 2nd generation ALK-TKI is being referred to here?

Thank you, specified.

  1. Page 7, line 314 - 317, cannot understand the phrase 'These data of important central efficacy highlight that, .........

Thank you, we better clarified the concept.

  1. Page 7, line 337, please elaborate on what was meant by putting crizotinib on the corner?

Thank you, we better clarified the concept.

  1. Table 2, please define what is frontline?

We replaced frontline with ‘first-line’

  1. Page 9, line 381, please elaborate on how re-biopsy and molecular profiling will guide subsequent treatment, in case of ALK tumor, or tumors with other targets?

Thank you, we better clarified the concept.

  1. Page 9, line 384 - 386, sentence meaning unclear, please elaborate further or re-write this part.

Thank you, we re-wrote it.

Minor concerns:

  1. Page 3, line 115, Different from .......
  2. Page 5, line 189, LAT 'was proven' to be safe......
  3. Page 6, line 269, please unify style of reporting p values.
  4. Page 9, line 345, each TKI has, indeed, 'its own unique' side effect profile.....
  5. Figure 1, 'addominal' should be 'abdominal'
  6. Line 413, use SI unit and standard abbreviations for 'mq' and same for subsequent appearance of 'mq'
  7. Line 430, /die?

Thank you, we have corrected all the typos indicated.

Reviewer 4 Report

In the present study, Chiara et al. provide an updated and comprehensive overview of the main treatment strategies in case of ALK rearranged NSCLC oligoprogression, including systemic treatment as well as local therapy. They also report two real-world clinical stories that prove the importance of multidisciplinary involvement and discussion in oncogene addicted patients with an oligoprogressive disease.

This is an important clinical topic. But I make a few suggestions :

  1.  In the section 2 titled “Oligoprogression in oncogene-addicted disease: focus on ALK-rearranged NSCLC” , concept and definition of oligometastatic disease  and oligoprogressive disease are However, data about oligoprogression in the ALK-rearranged NSCLC should also be provided. In addition, the description of CT and PET/CT feels redundant in this section.
  2.  In the section 3, “Despite these preclinical heterogeneous data, as a matter of fact, local therapies (including radiotherapy or surgery approaches) combined with targeted therapy prosecution is a current common habit”. The description of this sentence is not professional, I suggest another way to state it.
  3. Selected studies listed in Table 1 should be consistent with the studies presented in the section 3, or it will be confused.
       In the section 4, a table summarizing the cranial efficacy of different generation ALK-TKIs and local therapy is needed. Furthermore, Although CNS metastasis are also oligometastatic sites, it seems a little bit confusing to describe CNS separately.
  4. Section 5. “Systemic treatment algorithm: waiting for a guide”. I suggest to move this section to the beginning of the review, maybe as the section 2 after “introduction”, which feels more logical. Since the title of this review is “Management of oligoprogression in patients with metastatic NSCLC harboring ALK rearrangements.”
  5. Section 6, the case 2 seems to be redundant. Case 1 is enough.
  6. Table 3. Main ongoing clinical trials in oligoprogressive NSCLC. I suggest to put this part in the main text but not in the “Perspectives and conclusion” 

    Overall this is a very meaningful topic. However - I think the logic of the whole article needs to be improved, and certain sections could have more speculation on both the quality and comparability of the papers, as well as taking the findings together to give an overall conclusion.

Author Response

We thank the Reviewer for the comments. We implemented/modified our manuscript as suggested in order to better clarify all the important aspects highlighted.

This is an important clinical topic. But I make a few suggestions :

  1. In the section 2 titled “Oligoprogression in oncogene-addicted disease: focus on ALK-rearranged NSCLC”, concept and definition of oligometastatic disease and oligoprogressive disease are however, data about oligoprogression in the ALK-rearranged NSCLC should also be provided. In addition, the description of CT and PET/CT feels redundant in this section.

We thank the Reviewer for the comment. We modified the paragraph as suggested by eliminating the detailed description of PET/CT. We stressed the concept of oligoprogression management in patients harboring driver alterations. All data available for the ALK+ subgroup are reported. Unfortunately they are limited and, in many cases, heterogeneous.

  1.  In the section 3, “Despite these preclinical heterogeneous data, as a matter of fact, local therapies (including radiotherapy or surgery approaches) combined with targeted therapy prosecution is a current common habit”. The description of this sentence is not professional, I suggest another way to state it.

We thank the Reviewer for the comment. We reformulated the concept.

  1. Selected studies listed in Table 1 should be consistent with the studies presented in the section 3, or it will be confused.
       In the section 4, a table summarizing the cranial efficacy of different generation ALK-TKIs and local therapy is needed. Furthermore, Although CNS metastasis are also oligometastatic sites, it seems a little bit confusing to describe CNS separately.

We thank the Reviewer.

We modified table 1 as suggested.

In table 3 we have reported available data about CNS efficacy of different ALK-TKIs in terms of intracranial response rate. Available data on efficacy of different ALK-TKIs and local therapy are limited and for now heterogeneous and hardly comparable.

  1. Section 5. “Systemic treatment algorithm: waiting for a guide”. I suggest to move this section to the beginning of the review, maybe as the section 2 after “introduction”, which feels more logical. Since the title of this review is “Management of oligoprogression in patients with metastatic NSCLC harboring ALK rearrangements.”

We thank the reviewer for the suggestion. We implemented introduction and section two in order to improve the fluency of the manuscript. If it sounds fluid, we prefer to keep the original sequence in order to make the concept of oligoprogression central (more than treatment sequencing).

  1. Section 6, the case 2 seems to be redundant. Case 1 is enough.

Thank you, we removed Case 2.

  1. Table 3. Main ongoing clinical trials in oligoprogressive NSCLC. I suggest to put this part in the main text but not in the “Perspectives and conclusion” 

Thank you, we put the table in section 3.

Overall this is a very meaningful topic. However - I think the logic of the whole article needs to be improved, and certain sections could have more speculation on both the quality and comparability of the papers, as well as taking the findings together to give an overall conclusion.

We thank for the opportunity to improve our manuscript and we hope that it now results more fluid and clearer.

Round 2

Reviewer 3 Report

The authors have addressed most of the reviewer's queries.

One minor issue is the unit of drug dosing in the case report, it should either be mg / day or mg daily, but not mg / daily.

Reviewer 4 Report

Your revisions adequately address most of my previous major comments, thank you.